# The Genetic Landscape of Antimicrobial Resistance Genes in *Enterococcus cecorum* Broiler Isolates

**DOI:** 10.3390/antibiotics13050409

**Published:** 2024-04-29

**Authors:** Yue Huang, Filip Boyen, Gunther Antonissen, Nick Vereecke, Filip Van Immerseel

**Affiliations:** 1Department of Pathobiology, Pharmacology and Zoological Medicine, Faculty of Veterinary Medicine, Ghent University, Salisburylaan 133, 9820 Merelbeke, Belgium; yue.huang@ugent.be (Y.H.); gunther.antonissen@ugent.be (G.A.); 2Department of Translational Physiology, Infectiology and Public Health, Faculty of Veterinary Medicine, Ghent University, Salisburylaan 133, 9820 Merelbeke, Belgium; nickvereecke@hotmail.com; 3PathoSense BV, 2500 Lier, Belgium

**Keywords:** *Enterococcus cecorum*, antimicrobial resistance, antimicrobial resistance genes, penicillin-binding proteins, single nucleotide polymorphisms

## Abstract

*Enterococcus cecorum* is associated with bacterial chondronecrosis with osteomyelitis (BCO) in broilers. Prophylactic treatment with antimicrobials is common in the poultry industry, and, in the case of outbreaks, antimicrobial treatment is needed. In this study, the minimum inhibitory concentrations (MICs) and epidemiological cutoff (ECOFF) values (CO_WT_) for ten antimicrobials were determined in a collection of *E. cecorum* strains. Whole-genome sequencing data were analyzed for a selection of these *E. cecorum* strains to identify resistance determinants involved in the observed phenotypes. Wild-type and non-wild-type isolates were observed for the investigated antimicrobial agents. Several antimicrobial resistance genes (ARGs) were detected in the isolates, linking phenotypes with genotypes for the resistance to vancomycin, tetracycline, lincomycin, spectinomycin, and tylosin. These detected resistance genes were located on mobile genetic elements (MGEs). Point mutations were found in isolates with a non-wild-type phenotype for enrofloxacin and ampicillin/ceftiofur. Isolates showing non-wild-type phenotypes for enrofloxacin had point mutations within the GyrA, GyrB, and ParC proteins, while five amino acid changes in penicillin-binding proteins (PBP2x superfamily) were observed in non-wild-type phenotypes for the tested β-lactam antimicrobials. This study is one of the first that describes the genetic landscape of ARGs within MGEs in *E. cecorum*, in association with phenotypical resistance determination.

## 1. Introduction

*Enterococcus cecorum* is a facultative anaerobic, Gram-positive bacterium that is commonly found in the chicken gut. This bacterium can cause bacterial chondronecrosis with osteomyelitis (BCO), an emerging disease in broilers [1,2]. The disease is characterized by bacterial translocation from the gut to the bloodstream, followed by the hematogenous spread to internal organs, including tibiae, femora, and the caudal thoracic vertebrae [3,4]. Affected birds suffer from progressive lameness, which increases mortality due to sepsis or a lack of feed or water uptake [5].

While biosecurity and good management practices are key in preventing BCO, often, antimicrobial treatment is essential when animals develop disease. The pathogenesis is not fully understood but translocation to the bloodstream because of increased intestinal permeability seems an important first step. It has been shown that the loss of the intestinal barrier integrity, for example, due to heat stress, increases the translocation of *E. cecorum* [6]. Moreover, the use of lincomycin–spectinomycin on the first day post-hatching, a period of life in which the intestinal permeability is still high, can prevent BCO outbreaks [7]. When outbreaks occur, therapeutic antimicrobials are used, including tetracyclines and amoxycillin [8,9]. While high levels of resistance have been described against tetracyclines in various studies, amoxicillin resistance is generally lower. The latter antimicrobial is not a first-choice treatment for BCO, because of its importance in humans, and its frequent use may increase the risk of antimicrobial-resistant populations, including extended-spectrum β-lactamase-producing *Escherichia coli* (ESBL-producing *E. coli*) [10,11,12]. In general, only scarce data are available on antimicrobial resistance (AMR) in *E. cecorum* [10,11,13,14,15], and only few studies have investigated the molecular basis of the acquired resistance, i.e., the determination of related antimicrobial resistance genes (ARGs) or single nucleotide polymorphisms (SNPs) [15]. Whole-genome sequencing (WGS) has become a useful tool to identify bacterial ARGs, and SNPs in many bacterial species [15,16,17,18,19]. It can also be used to identify mobile genetic elements (MGEs) such as insertion sequences (ISs), transposons (Tns), plasmids, integrons (Ins), and integrative conjugative elements (ICEs). All playing a critical role in the processes of capturing, accumulating, and disseminating ARGs, amongst others [20,21].

In the current study, *E. cecorum* strains isolated from diseased and healthy chickens were selected based on their pulsotype, as previously determined [22]. On these strains, broth microdilution assays and Oxford Nanopore Technologies long-read WGS were performed. We evaluated (i) the minimum inhibitory concentration (MIC) phenotypic distributions for a selection of antimicrobials (*n* = 10), (ii) the relationship between these AMR phenotypes and genotypes, and (iii) the location of ARGs on MGEs.

## 2. Results

### 2.1. Determination of MIC and Epidemiological Cutoff (ECOFF) Values (CO_WT_)

The results of the four quality control strains were within the acceptable quality control ranges as described by the Clinical and Laboratory Standards Institute (CLSI, 2023) [23]. The distributions of the MIC values for the ten tested antimicrobials are shown in Table 1 and the MIC data for all 53 *E. cecorum* strains are shown in Appendix A.

Typical bimodal or even multimodal MIC distributions were observed (Figure 1), pointing to acquired resistance against all antimicrobials, except for gentamicin. The cutoff (CO_WT_) values were calculated for the ten antimicrobial agents based on the data from the 53 *E. cecorum* strains (Table 1, Figure 2, and Appendix A). A high proportion of non-wild-type strains for tetracycline was observed (68.9% for clinical strains and 100% for non-clinical strains). The proportions of non-wild-type strains for lincomycin, spectinomycin, and the lincomycin–spectinomycin combination were also high: lincomycin: 42.2% for clinical strains and 100% for non-clinical strains, spectinomycin: 37.8% for clinical strains and 100% for non-clinical strains, and lincomycin–spectinomycin (1:2): 42.2% for clinical strains and 100% for non-clinical strains. There are no indications of cross-resistance between both tested aminoglycosides (gentamicin and spectinomycin), as no gentamicin-acquired resistance was observed, in contrast to spectinomycin. Cross-resistance was observed between both tested β-lactam antimicrobials, ampicillin and ceftiofur, as all strains classified as the non-wild-type phenotype for ceftiofur also showed an acquired resistance for ampicillin. One strain showing ampicillin-acquired resistance was also borderline, categorized to belong to the wild-type phenotype for ceftiofur by the normalized resistance interpretation method (NRI). All strains showing tylosin-acquired resistance also showed lincomycin resistance, while few clinical strains showed lincomycin resistance, but no tylosin-acquired resistance. Importantly, acquired resistance was observed for the critically important antimicrobial agents enrofloxacin and vancomycin, though mainly in non-clinical strains. Relatively high MIC values (8–32 µg/mL) for strains belonging to the wild-type populations of gentamicin and spectinomycin suggest that aminoglycosides are only moderately active against *E. cecorum*.

### 2.2. Identification of Resistance Genes

A total of 14 strains were subjected to WGS using long-read nanopore sequencing. Genome assembly sizes ranged between 2,312,828 and 3,086,325 bp, with all assemblies showing a 94.34 ± 1.40% completeness based on CheckM completeness using an *Enterococcus* spp. database with 672 marker genes from 51 *Enterococcus* genomes. All were confirmed to be *E. cecorum* using rMLST. The genome assemblies showed 36.6 ± 0.1 GC% and 2853 ± 331 annotated coding sequences (CDSs) (Appendix A). Based on the SNP-based phylogenetic maximum-likelihood inference, clinical and non-clinical isolates clustered together (Figure 3). The ARGs were identified in the genomes of 14 strains, which showed a clear association with the observed AMR phenotypes as presented in Figure 3.

In accordance with the phenotypic susceptibility results, all selected non-clinical isolates (*n* = 8) carried the aminoglycoside resistance gene *ant(6)-la*, the macrolide (tylosin) resistance gene *ermB* and the lincosamide (lincomycin) resistance gene *lsaE*. All these strains also carried the *lsaB* gene, which only showed a 64.76 ± 0.15% nucleotide identity (% nt. ID). Of note, also, the genes *emeA* (*n* = 1; 66.90% nt. ID), *mel* (*n* = 4; 89.24 ± 0.12% nt. ID), and *msrE* (*n* = 4; 64.40 ± 0.03% nt. ID) were identified in some strains (Appendix A). All selected non-clinical strains also carried at least one tetracycline-associated ARG, with all of them containing the *tet(M)* gene and five of them containing the *tet(L)* gene. Four out of eight non-clinical strains additionally carried the glycopeptide (vancomycin) resistance gene operon (*vanA*, *vanHA*, *vanRA*, *vanSA*, *vanXA*, *vanYA*, and *vanZA*). In contrast, only two out of six clinical isolates carried *ant(6)-la* and *ermB* genes, and four out of six carried at least one tetracycline resistance gene, with four of them containing the *tet(M)* gene, and two of them containing the *tet(L)* gene and *tet(O)* gene. As expected, all of our resistance-gene-positive isolates were non-wild-type based on the CO_WT_ values. The *lsaE* gene was not detected in the two clinical strains that exhibited a non-wild-type phenotype for lincomycin resistance; however, the *ermB* gene, which also contributes to lincomycin resistance, was detected in these two clinical strains.

### 2.3. Mutations in the E. cecorum GyrA/GyrB/ParC and Genes Encoding Penicillin-Binding Proteins (PBPs) Are Associated with Enrofloxacin and β-Lactam Antimicrobial Resistance

Two out of eight non-clinical isolates were non-wild-type for enrofloxacin, while six out of eight non-clinical isolates were non-wild-type for both ampicillin and ceftiofur. While no known ARGs, as presented in the used Comprehensive Antimicrobial Resistance Database (CARD), were identified to be linked to these observed AMR phenotypes, we analyzed the genome sequences for chromosomal SNPs in the genes encoding quinolone-resistance-determining regions (QRDRs) and penicillin-binding proteins (PBPs) that may explain the resistance to fluoroquinolones and β-lactam antimicrobials, respectively (Figure 3 and Figure 4). For the enrofloxacin non-wild-type strains, mutations within the QRDRs were present in the GyrA, GyrB, and ParC protein (Ser83Tyr, Thr377Met, and Asn529Asp, respectively). A total of five putative PBPs were identified in all selected *E. cecorum* strains, including PBPs belonging to the PBP1a superfamily 1, PBP1a superfamily 2, PBP1a superfamily 3, PBP2 superfamily, and PBP2x superfamily as presented in Figure 4A. The data showed that all six non-wild-type strains (*E. cecrum* 33, 65, 66, 67, 123, and 127) for both ampicillin and ceftiofur exhibited five point mutations at positions in the active domain site of the transpeptidase of the PBP2x superfamily PBP protein, including Thr335Ile, Met346Thr, Gln366Lys, Trp377Ser, and Leu382Ile (Figure 4B). A predicted (AlphaFold) protein structure of the PBP2x superfamily highlights its PBP-like structure and the location of the amino acid changes within the transpeptidase active domain (Figure 4C).

### 2.4. Mobile Genetic Elements Associated with AMR

An analysis of the whole-genome sequencing data showed that all detected resistance genes are located on MGEs, including ISs and Tns (Figure 5, Figure 6 and Figure 7). The *van* gene landscape showed that all *van* operon-positive strains showed the presence of the IS*1380* family transposase, located upstream of the *vanRA* gene, which contributes to the acquisition of ARGs (Figure 5A). The Tn*1546* resolvase and Tn*3* family transposase were detected in three out of four *van* operon-positive strains, adjacent to an IS*1380* family transposase. In particular, Tn*1546* has been reported as being responsible for vancomycin resistance among enterococcal species [20]. Four of the *ant(6)-la*-positive strains also showed the presence of an IS*1380* family transposase, located upstream and downstream of the *ant(6)-la* gene. For the rest of *ant(6)-la*-positive strains, an IS*L3* family transposase was detected, which was also found near the *lsaE* and *lsaB* genes (Figure 5B). Of note, *E. cecorum* strain 33 also harbored both *lsaE* and *lsaB* genes, which were located further upstream of the *ant(6)-Ia* region. The transposon Tn*917* resolvase, associated with the resistance to macrolide–lincosamide–streptogramin (MSL) antimicrobials, was found in all ten *ermB*-positive strains, adjacent to *ermB* (Figure 6). The *tet(M)* and *tet(L)* gene landscape showed the presence of a Tn*1545* transposase in all tetracycline-resistant strains, located downstream of the *tet(M)*/*tet(L)* gene (Figure 7). It has been previously described that the Tn*1545* transposon carries the ARGs *tet(M)*, *ermB*, and *aphA-3* [24,25].

## 3. Discussion

BCO has emerged as a major concern in broilers, and unraveling the pathogenesis of this disease is crucial for resolving this problem. Antimicrobial drugs have been used prophylactically and therapeutically and concerns have been raised about the acquisition of AMR. Therefore, in this study, the AMR profile of 53 *E. cecorum* isolates from BCO outbreaks and healthy poultry were analyzed. Even though this collection of strains allowed an in-depth investigation of the genetic background of the phenotypically observed resistance, the relatively small number of strains, and, especially, of non-clinical strains, is a limitation of the current study, possibly affecting certain outcome parameters such as the percentages of resistance or the ECOFFs.

In our study, all non-clinical and 26.7% of clinical *E. cecorum* isolates exhibited a resistance to macrolides, with tylosin resistance mediated by the *ermB* resistance gene in the non-wild-type strains. Resistance to macrolides in enterococci from both animal and human sources has been reported [26,27,28]. In previous studies, acquired resistance to the macrolides could be attributed to the presence of one or more resistance genes, such as erythromycin resistance methylase (*erm*) genes (*ermB*, *ermC*, and *ermG*), and efflux genes responsible for the pumping of the antimicrobial from the cell (*mefA*, and *msrD*), of which the *ermB* gene dominated among the enterococci [15,28]. Moreover, the *ermB* gene confers resistance to MSLtype B, and is associated with transposons on chromosomes or plasmids [29]. A previous study showed that the dissemination of the *ermB* gene can occur via transposable elements on chromosomes rather than plasmids [30]. Another study demonstrated that the transposon Tn*917* is the main carrier of the *ermB* gene in *S. pneumoniae* isolates [31]. Not surprisingly, all *ermB* gene-positive *E. cecorum* strains in our study harbored the transposon Tn*917* located adjacent to the *ermB* gene, confirming the Tn*917* transposon as its main carrier.

For resistance to lincomycin/spectinomycin, several studies showed that commensal isolates of *E. cecorum* exhibited more resistance to lincomycin and spectinomycin compared to pathogenic isolates, which was similar in our study. Of note, most studies, including our study, comprised a low number of non-clinical isolates [11,13,15]. The combinatorial treatment of lincomycin–spectinomycin in newly hatched birds has been a successful strategy in preventing *E. cecorum*-associated diseases. Comparing the lincomycin, spectinomycin, and the lincomycin–spectinomycin (1:2) combination MIC results, it can be concluded that *E. cecorum* is only moderately susceptible to spectinomycin, with the wild-type population showing MIC values in the range of 16–32 µg/mL. Adding spectinomycin does not seem to have a clear added value when used in combination with lincomycin both for strains showing wild-type or non-wild-type properties for lincomycin. Moreover, the preventive use of antimicrobials in groups of animals has been banned in the EU since 2022. In our study, all eight non-clinical and two clinical isolates were found to harbor *ant(6)-la* genes, along with the *lsaE* or *ermB* gene. Additionally, the *lsaB* gene was identified in all (*n* = 8) non-clinical strains, showing a low 64.76 ± 0.15% nt. ID as compared to the *Mammaliicoccus sciuri lsaB* reference sequence (AJ579365.1). Many of these genes are associated with MGEs located in the chromosome. As an example, the IS*1380* family transposase seems to have captured various ARGs including *rmtC* (aminoglycoside resistance), *qnrE1* (fluoroquinolone resistance), and *bla_OXA-204_* (cephalosporin resistance), and it has been reported that the IS*1380* family IS*Ecp1*-like element is associated in the gentamicin resistance gene transfer in *E. casseliflavus* [20,32].

Tetracycline resistance was conferred by genes *tet(M)*, *tet(O)*, and *tet(L)* in both clinical and non-clinical *E. cecorum* strains. Other reports suggested that a high prevalence of tetracycline-resistant *E. faecalis* were found in poultry, with 35% and 55% of tetracycline-resistant *E. faecalis* in chicken and other poultry meat, respectively [33]. The resistance of enterococci to tetracycline has been associated with the therapeutic use of this antimicrobial on farms, and plasmids conferring this resistance have been reported [34]. In *Enterococcus*, ARGs linked to ribosomal protection (*tet(M)*, *tet(O)*, and *tet(S)*) and enzymatic inactivation (*tet(K)* and *tet(L)*) are commonly found antimicrobial resistance mechanisms [34]. Among these, *tet(M)* is most frequently observed and is usually found in close proximity to the *ermB* gene [15,29]. In our data, ten out of twelve *tet(M)*-positive *E. cecorum* isolates also harbored the *ermB* gene, supporting this. In addition, all *tet(M)*-positive *E. cecorum* isolates carried the Tn*1545* transposon. According to previous research, the *tet(M)* gene is most frequently located on the chromosome, and this ARG can be carried by the Tn*916*/Tn*1545* family of conjugative transposons [35]. These transposons play a crucial role in the conjugative transfer and transposition in both Gram-negative and Gram-positive bacteria, including enterococci, as the Tn*916*/Tn*1545* family is very prevalent [36,37].

Fifty percent and 4.4% of *E. cecorum* isolates from non-clinical and clinical strains appear to be resistant to vancomycin, respectively. Three out of four vancomycin resistance gene clusters were found to be adjacent to a Tn*1546* element [38]. In the European Union (EU) and Asia, glycopeptides were used in poultry production as a growth promoter, such as the vancomycin analog avoparcin [39,40]. However, its use as a feed additive has been banned since 1997 [41,42]. Hence, vancomycin resistance in *E. cecorum* isolates has been seldom found. Based on previous studies, a *vanA*-gene-positive *E. cecorum* strain was found in September 2009 [14], and, also, one vancomycin resistance broiler isolate was reported in 2016 [43]. According to the literature, *E. cecorum* could serve as a reservoir of vancomycin resistance genes and can be part of the vancomycin-resistant enterococci (VRE) [14,44]. Therefore, conducting surveillance on the VRE type of *E. cecorum* is essential, as it may pose a potential risk of AMR for humans through the food chain.

Multiple amino acid substitutions within the quinolone-resistance-determining regions (QRDRs) of both the DNA gyrase (GyrA and GyrB) and topoisomerase IV (ParC), and penicillin-binding proteins (PBPs) were identified. These mutations correlated with significantly higher MIC values compared to *E. cecorum* strains that did not harbor these mutations. Based on previous studies, PBP 5 and PBP 4 were often found in *E. faecalis* and *E. faecium*, and had a low affinity for penicillin [45,46,47]. However, in another study, the PBP types in *E. cecorum* might be more closely related to *Streptococcus* species rather than other *Enterococcus* species [48]. In our study, it was shown that all the point mutations were found in the PBP2x superfamily. This type of PBPs (PBP2x) was often found in *Streptococcus pneumoniae*, playing an important role in β-lactam antimicrobial resistance [49,50,51]. Here, we report five new *E. cecorum* PBP2x amino acid changes (Thr335Ile, Met346Thr, Gln366Lys, Trp377Ser, and Leu382Ile) within the PBP transpeptidase active domain, potentially contributing to ampicillin and ceftiofur resistance.

We also describe amino acid changes associated with enrofloxacin resistance in *E. cecorum*. The GyrA protein mutation (Ser83) has been previously observed in *Mycoplasmopsis bovis* (formerly known as *Mycoplasma bovis*) and *Salmonella enterica*, where it is associated with enrofloxacin and ciprofloxacin resistance [18,52,53]. On the other hand, the reported mutations in GyrB and ParC are not-yet-described mutations in *E. cecorum* associated with fluoroquinolone resistance, which might require further validation to link them with actual resistance phenotypes.

Current findings suggest that acquired resistance towards the major antimicrobial classes used to control BCO can be observed in *E. cecorum*. However, as there are currently no clinical breakpoints for *E. cecorum* infections in poultry, the clinical interpretation of these results is not straightforward. It can be assumed that strains with acquired resistance showing MIC values 10–100 times higher than the wild-type population may not be efficiently treated with these antimicrobial agents. As the observed ARGs are strongly linked with MGE, it is tempting to speculate that these resistance genes are horizontally acquired and can further spread horizontally, though it is not clear yet which bacterial species might be involved in such events.

Summarized, AMR in broiler *E. cecorum* isolates was relatively high, even against critically important agents such as vancomycin. Genes conferring resistance were often found on MGEs, while fluoroquinolone and β-lactam antimicrobial resistance were found to be mediated by SNPs and were characterized in detail for the first time in *E. cecorum*. Therefore, the current results contribute to an understanding of the genetic landscape of ARGs in *E. cecorum* and provide a basis for further research to understand the mechanism of the antimicrobial resistance epidemiology in *E. cecorum*.

## 4. Materials and Methods

### 4.1. Bacterial Strains

*E. cecorum* strains used in this study (*n* = 53) were isolated from healthy (non-clinical strains, *n* = 8) and BCO-affected chickens (clinical strains, *n* = 45) in Belgium during 2019–2020. Strain characteristics have been described previously (source of isolation, PFGE type, and presence of virulence genes) [22]. All *E. cecorum* isolates were cultured on Columbia sheep blood agar plates and incubated at 37 °C in a 5% CO_2_ atmosphere. Reference strains *Enterococcus faecalis* ATCC 29212, *Staphylococcus aureus* ATCC 29213, *Escherichia coli* ATCC 25922, and *Streptococcus pneumoniae* ATCC 49619 were used as quality control strains in the susceptibility tests and were cultured on Columbia sheep blood agar plates and incubated at 37 °C in a 5% CO_2_ atmosphere.

### 4.2. Antimicrobial Susceptibility Testing

Antimicrobial susceptibility testing was carried out for ten antimicrobial agents: ampicillin, vancomycin, lincomycin, spectinomycin, lincomycin–spectinomycin, tetracycline, tylosin, gentamicin, enrofloxacin, and ceftiofur (all obtained from Sigma-Aldrich, Darmstadt, Germany). Due to the fastidious growth characteristics of *E. cecorum*, that more closely resemble *Streptococcus* rather than *Enterococcus* characteristics, the broth microdilution method as described for *Streptococcus* spp. in the Clinical Laboratory Standards Institute (CLSI, 2023) standards was used [23]. The antimicrobials were dissolved in an appropriate solvent, and then further diluted in sterile distilled water. The Minimum Inhibitory Concentration (MIC) was determined using Mueller–Hinton П agar (Becton Dickinson, Cockeysville, MD, USA) supplemented with 5% sheep blood and incubated at 37 °C in a 5% CO_2_ atmosphere, containing two-fold dilutions of the antimicrobials. Concentrations of the antimicrobials ranged from 0.03 μg/mL to 128 μg/mL, except for the combination lincomycin–spectinomycin (1:2), which was tested in a range between 0.03/0.06 µg/mL and 128/256 µg/mL, respectively. Uninoculated antimicrobial free agar plates were included as sterility controls, and inoculated antimicrobial free agar plates were included as growth controls. Inoculated plates were prepared by suspending colonies from an overnight grown culture on Columbia blood agar (Thermo Fisher Scientific, Merelbeke, Belgium), in buffered saline, to a density of 0.5 McFarland standards. Approximately 1 × 10^7^ colony-forming units of each strain were inoculated on the plates. Resulting MIC values were recorded after incubation at 37 °C in a 5% CO_2_ atmosphere for 24 h and were defined as the lowest concentration producing no visible growth. *Enterococcus faecalis* ATCC 29212, *Staphylococcus aureus* ATCC 29213, *Escherichia coli* ATCC 25922, and *Streptococcus pneumoniae* ATCC 49619 were used as quality controls.

### 4.3. ECOFF Values Determination

As no EUCAST ECOFFs are available for *E. cecorum*, wild-type cutoff values (CO_WT_) were determined using the “Normalized Resistance Interpretation (NRI)” method (Bioscand AB, Täby, Sweden) [54], as a proxy for the ECOFF. The outcome is limited to a tentative estimate of the ECOFF when the standard deviation of wild-type MIC values in the normal distribution exceeds 1.2 log2.

### 4.4. DNA Extraction and Whole-Genome Sequencing

Fourteen *E. cecorum* strains were selected according to the MIC results of different antimicrobial agents, ensuring strains were included to cover all types of resistance to the selected antimicrobials. All strains were grown as described above and an overnight culture was subjected to high-molecular-weight DNA extraction and native long-read DNA sequencing using the Oxford Nanopore Technologies GridION system (Oxford, UK) as described before [21,55,56]. In short, DNA was extracted using the ZymoBIOMICS DNA MiniPrep kit (ZymoResearch, Irvine, CA, USA), with the addition of a Proteinase K treatment (20 µg∙µL^−1^; Promega, Madison, WI, USA). Low-quality DNA samples were subjected to an extra DNA clean-up using CleanNGS (CleanNA, Waddinxveen, The Netherlands) magnetic beads at a 1:1 ratio. The SQK-RBK004 (ONT) quick library preparation was used to multiplex up to 10 strains on a single MinION flow cell. Raw data were collected in MinKNOW (ONT) and base called using the super accurate model in guppy (v6.3.9; ONT). An overview of sequencing throughput and statistics is given in Appendix A. Raw data were used to assemble genomes as described in Vereecke et al., 2023 [21] using Trycycler (v.0.5.3; [57]), minimap2 (v2.20; [58]), and medaka (v.1.7.3; ONT). All software were run at default settings as depicted on the Trycycler wiki (https://github.com/rrwick/Trycycler/wiki, accessed on 27 January 2023). Final genomes were quality checked using CheckM (v1.1.0; [59]), including 672 markers from 51 *Enterococcus* species, and classified up to the species level using rMLST on pubMLST [60]. An overview of assembly quality control, coverage, and accession numbers (BioProject: PRJNA1071719) can be found in Appendix A.

### 4.5. Phylogenetic Inference, Identification of ARGs, Mobile Genetic Elements, and Point Mutations

The resulting genome assemblies were used in an SNP-based phylogenetic maximum-likelihood (ML) inference using csi phylogeny [61] and IQtree (v.1.6.12; –bb 1000 -m GTR + R + I; [62]). Next, genome assemblies were screened for ARGs using Abricate (v.1.0.1; --minid 60 --mincov 80; https://github.com/tseemann/abricate, accessed on 27 January 2023) against the CARD database [63]. An overview of all identified ARGs is given in Appendix A. Final visualizations were made in iTOL (v.5; [64]). The characterization of MGEs was carried out by extracting flanking regions of ARGs using flanker (v.0.1.5; [65]), which were then annotated and visualized using Bakta (v.1.7.0; --db db-full --compliant --genus *Enterococcus* --species *cecorum* --gram +; [66]) and Clinker (v.0.0.26; [67]), all at default settings. Identification of point mutations was obtained by extracting target genes from the Bakta annotated genomes, aligning them with MAFFT (v.7.520; [68]), and manually annotated and/or translated in Mega11 (v.11.0.11; [69]). For PBP proteins, protein homology and structures were determined using NCBI blastP and Interpro for the (sub)classification of protein families (https://blast.ncbi.nlm.nih.gov/Blast.cgi and https://www.ebi.ac.uk/interpro/search/sequence/, accessed on 30 November 2023 with default settings). The PBP protein structure prediction was made using the Alphafold2 google colab at preset settings [70].

## Figures and Tables

**Figure 1 antibiotics-13-00409-f001:**
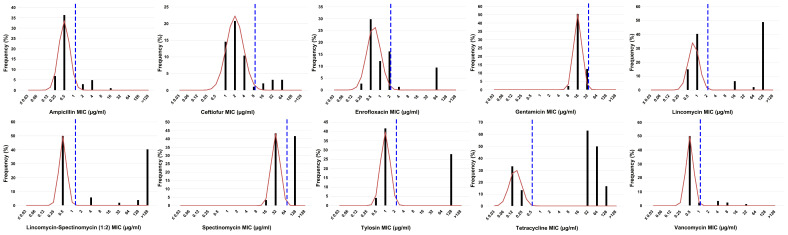
Typical bimodal and multimodal MIC distributions for ten antimicrobial agents on 53 *E. cecorum* isolates, except for gentamicin (unimodal MIC distribution). These plots are representing the results as shown in the automated NRI software. The red line indicates the wild-type population distribution as calculated by NRI software.The blue dotted line indicates the CO_WT_ value as proxy for the ECOFF.

**Figure 2 antibiotics-13-00409-f002:**
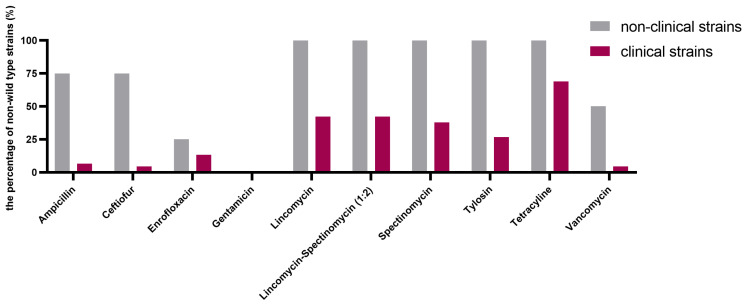
The percentage of non-wild-type *E. cecorum* strains of non-clinical (grey, *n* = 8) and clinical strains (purple, *n* = 45) according to the CO_WT_ values for ten antimicrobials.

**Figure 3 antibiotics-13-00409-f003:**
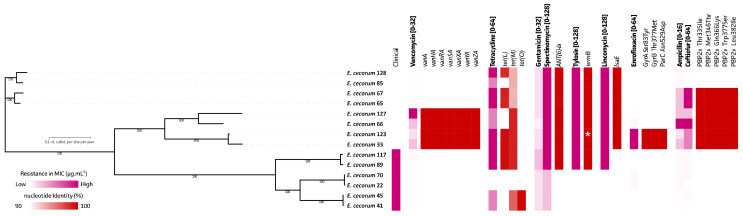
Presence of antimicrobial-resistance-associated genes in 14 *E. cecorum* strains (6 clinical and 8 non-clinical isolates) and the association with the resistance phenotype. The bold style of antimicrobials indicates the tested antimicrobials, their range of tested concentrations [ ], and their MIC values, which are color-scaled from low (white and light pink) to high (bright pink). A complete overview of all identified ARGs is shown in Appendix A. For the enrofloxacin non-wild-type strains, mutations within the quinolone-resistance-determining regions (QRDRs) were present in the GyrA, GyrB, and ParC proteins (Ser83Tyr, Thr377Met, and Asn529Asp, respectively). The non-wild-type strains for both ampicillin and ceftiofur contains five point mutations at the PBP2x superfamily PBP protein, including Thr335Ile, Met346Thr, Gln366Lys, Trp377Ser, and Leu382Ile. *: *E. cecorum* 123 contains two *ermB* genes. Bootstrap values are represented as percentages for each node within the SNP-based maximum-likelihood phylogenetic tree.

**Figure 4 antibiotics-13-00409-f004:**
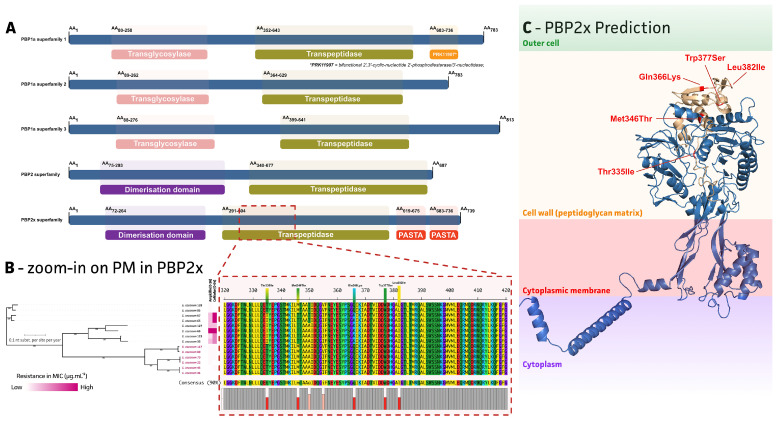
The putative penicillin-binding proteins (PBPs) in *E. cecorum* strains. (**A**) Five different types of PBPs were determined using blastP; (**B**) multiple point mutations were found in PBP2x superfamily for ampicillin- and ceftiofur-resistant isolates. The strain number of clinical isolates were colored purple; (**C**) a predicted (AlphaFold) protein structure of the PBP2x with five point mutations. The PASTA domain is also referred to as the penicillin-binding protein and serine/threonine kinase associated domain within PBPs.

**Figure 5 antibiotics-13-00409-f005:**
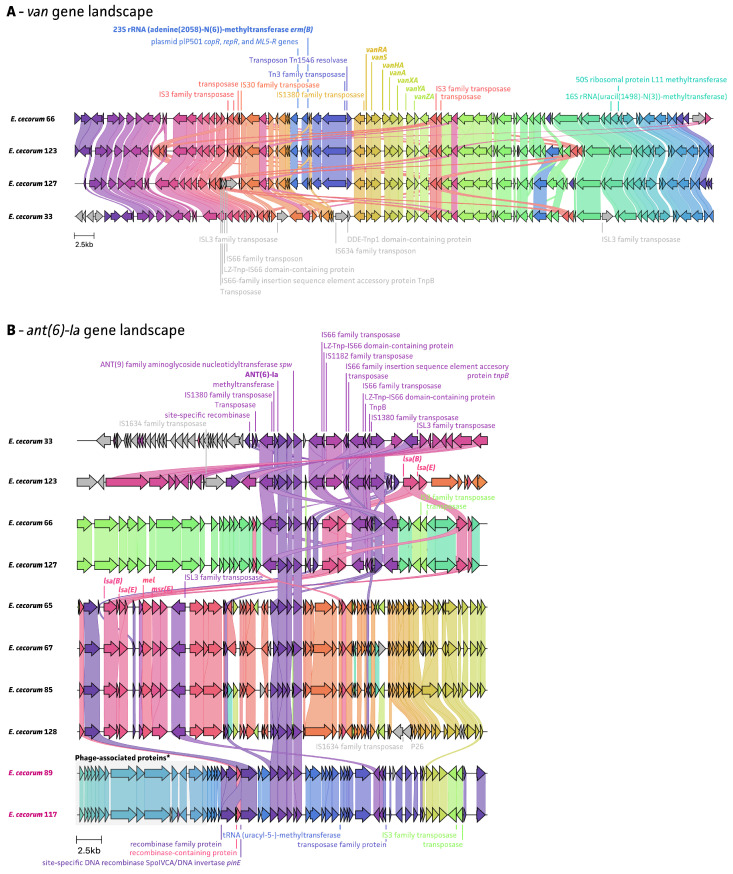
The landscape of antimicrobial-resistance-associated genes with MGEs. The strain number of clinical isolates were colored purple. (**A**) The *van* gene landscape for four non-clinical *E. cecorum* strains; (**B**) the *ant(6)-la* gene landscape for eight non-clinical *E. cecorum* strains and two clinical *E. cecorum* strains.

**Figure 6 antibiotics-13-00409-f006:**
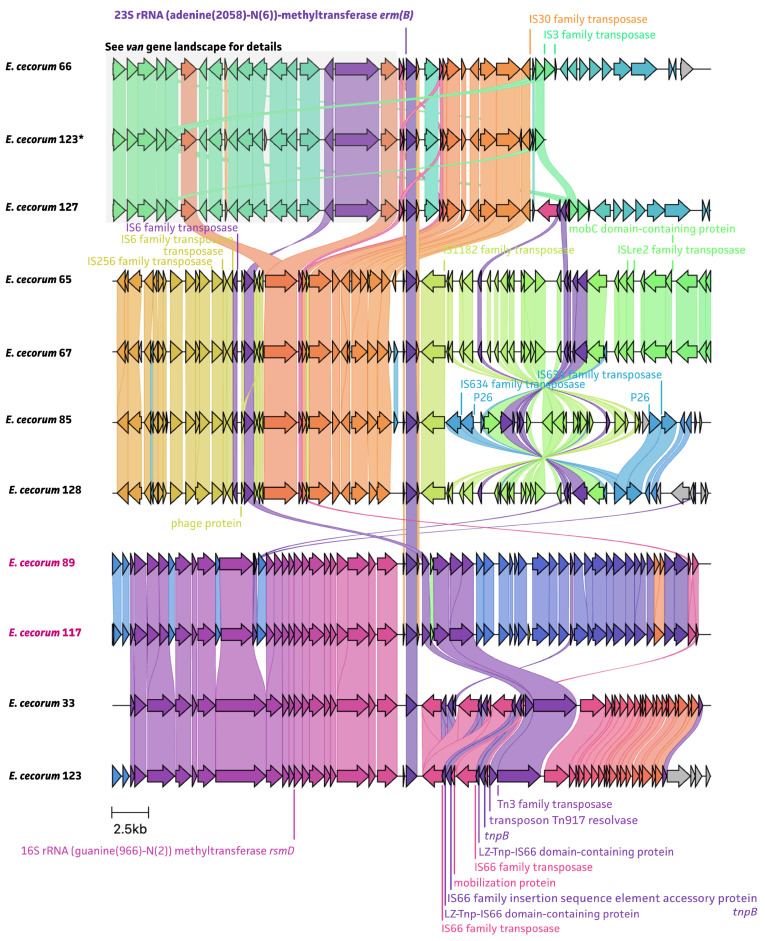
The landscape of antimicrobial-resistance-associated genes with mobile genetic elements of the *ermB* gene for eight non-clinical *E. cecorum* strains and two clinical *E. cecorum* strains, of which *E. cecorum* 123 were two hits for *ermB* gene in genome, and *E. cecorum* 123* means the second *ermB* gene.

**Figure 7 antibiotics-13-00409-f007:**
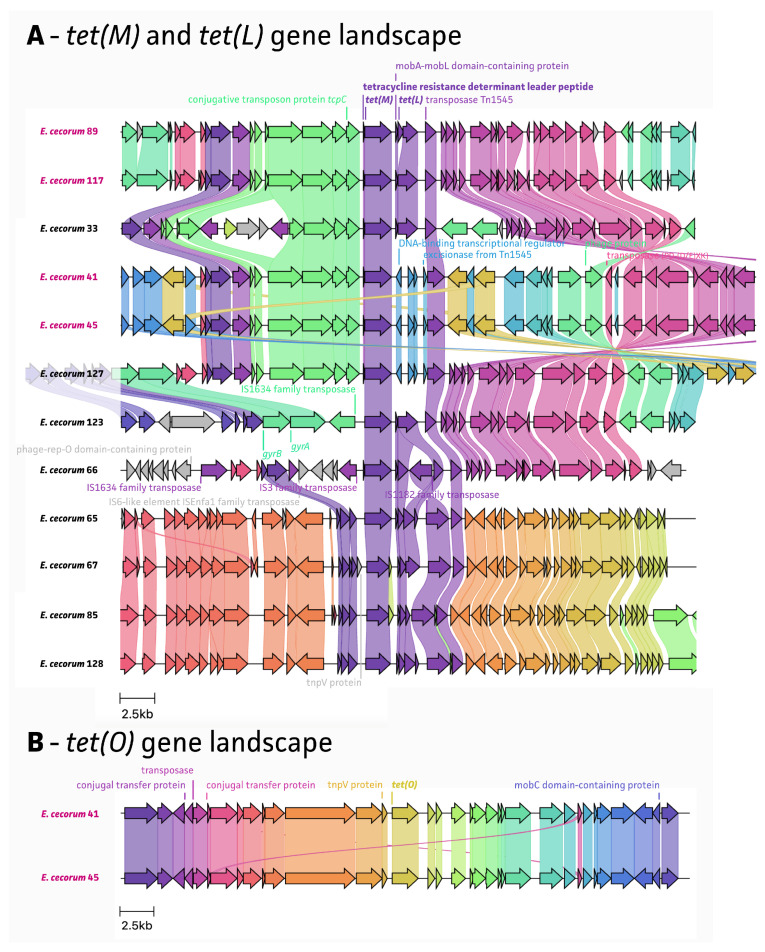
The landscape of antimicrobial-resistance-associated genes with mobile genetic elements for tetracycline resistance. (**A**) The *tet(M)*/*tet(L)* genes landscape for eight non-clinical *E. cecorum* strains and four clinical *E. cecorum* strains. (**B**) The *tet(O)* gene landscape which was only found in two clinical *E. cecorum* strains.

**Table 1 antibiotics-13-00409-t001:** Distribution of MIC values for ten different antimicrobial agents and the calculated ECOFF value (CO_WT_).

Antimicrobial Agent	Number of Strains with MIC (µg/mL)																	
															Wild-Type		Non-Wild-Type	
	≤0.03	0.06	0.12	0.25	0.5	1	2	4	8	16	32	64	128	>128	[*n*]	[%]	[*n*]	[%]
Ampicillin				7	37		3	5		1					44	83%	9	17%
Ceftiofur						14	20	10	1	2	3	3			45	85%	8	15%
Enrofloxacin				2	22	9	12	1				7			45	85%	8	15%
Gentamicin									2	40	11				53	100%	0	0
Lincomycin					7	19				3		1		23	26	49%	27	51%
Lincomycin–Spectinomycin (1:2)					26			3			1		2	21	26	49%	27	51%
Spectinomycin										2	26			25	28	53%	25	47%
Tylosin					3	30								20	33	62%	20	38%
Tetracycline			10	4							19	15	5		14	26%	39	74%
Vancomycin					45	2		3	2		1				47	89%	6	11%

A black vertical line indicates the CO_WT_ value, as calculated by the normalized resistance. interpretation (NRI) method. For lincomycin–spectinomycin (1:2), the antimicrobial concentrations in the table indicate the concentration of lincomycin. [*n*]: number of *E. cecorum* strains.

## Data Availability

Genomes were submitted to NCBI and are available under the BioProject PRJNA1071719 with accession numbers as presented in Appendix A.

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
