# Peer review of "The Genetic Landscape of Antimicrobial Resistance Genes in Enterococcus cecorum Broiler Isolates"

_antibiotics, 2024, doi:10.3390/antibiotics13050409_

Round 1

Reviewer 1 Report

Comments and Suggestions for Authors

The manuscript is well-written, and the data is clear and sound. There is only one big issue that came out to me in the results section:

The higher percentage of non-clinical strains may be due to the much smaller strain numbers (n=8) compared to the clinical strains (n=45). The authors may consider adding more non-clinical strains to make the results more reliable statistically.

Reviewer 2 Report

Comments and Suggestions for Authors

The manuscript presents a comprehensive study on E. cecorum associated with bacterial chondronecrosis with osteomyelitis (BCO) in broilers, emphasizing the importance of antimicrobial treatment. The study effectively links phenotypical resistance with genotypical factors, highlighting mobile genetic elements (MGEs) as carriers of antimicrobial resistance genes (ARGs). The research contributes to understanding the genetic landscape of ARGs and their phenotypical manifestations in E. cecorum, addressing an emerging concern in poultry health.

Here are several suggestions:

Line 15, the term "MIC" should be pluralized to "MICs".

Line 68, the abbreviation for epidemiological cutoff values should be ECOFF. Please revise the abbreviation throughout the manuscript for consistency.

Line 101, it would be better to include the bimodal and multimodal distribution graphs for a more visual representation of the data, which would greatly enhance the reader's understanding of the results.

Line 128, the criteria for selecting the 14 strains for whole genome sequencing (WGS) should be elucidated. Providing this information will add clarity to the methodological rigor of the study.

The manuscript would benefit from an expanded discussion regarding the practical implications of these findings, specifically how they might influence the management of BCO in broilers. Additionally, a succinct summary of the core findings and their implications for poultry health and antimicrobial resistance management, alongside future research directions, would provide a more comprehensive conclusion to the paper.

Questions for the Authors:

Could the authors provide an explanation for the presence of specific resistance genes in E. cecorum? Is there any substantiated evidence of horizontal gene transfer contributing to ARG dissemination within poultry farm environments?

Have the authors noted similar trends in antimicrobial resistance among E. cecorum or other bacteria in different countries or across other livestock species? Comparing these trends could offer valuable insights into the global patterns of ARG distribution.

Reviewer 3 Report

Comments and Suggestions for Authors

This ms reports AMR genes in Enterococcus cecorum isolates of chickens. ECVs were determined for 10 antimicrobials used in veterinary medicine or of critically important in human medicine. The approach used is straightforward. The findings help fill the data gaps in E. ceorum. The ms could be improved for better clarity. Comments are given below.

L21. “All” needs to be clarified. Perhaps, change to “These” to refer previous sentence, which, for example, does not include fluoroquinolones, even thought QRDRs are studied and are hardly considered to be associated with “mobile

L102/L351 (section 4.3).  How the method used (Ref 54) is compared with EUCAST’s SOP for ECOFF calculation: https://www.eucast.org/fileadmin/src/media/PDFs/EUCAST_files/EUCAST_SOPs/2021/EUCAST_SOP_10.2_MIC_distributions_and_epidemiological_cut-off_value__ECOFF__setting_20211202.pdf . As well, the numbers of strains used are limited and this limitation should be mentioned.

L110-111. Need clarity for “both tested aminoglycosides”: first , specify them. Likely refer to gentamicin and spectinomycin. Strictly speaking, only gentamicin is an aminoglycoside while spectinomycin is an aminoglycyclitol.

L127. There are only 14 strains that were done by WGS, which is a limitation to be mentioned.

L179. The strains numbers of 6 non-wild type should be included in the text it takes a while to note them in Fig 3B middle (but almost illegible) .

L318. Please include the place(s) and year(s) for the isolates obtained. The information helps understand the significance of the findings.

Minors (Examples)

L12/L32. Remove “(E. cecorum)” as it is note needed in microbiology writing.

L15/L68/Table 1 title. Write “(ECV)” (CLSI) or “ECOFF” (EUCAST) before  for “COWT” which is for wildtype cut-off”. Inf needed, introduce COWT (as noted in L102), an important parameter in CLSI document.

/23, etc. Strongly suggest for not using “PMs” as it is so infrequently used in the field and merely use full spelling for being  reader-friendly.

L26. Write “E. cecrum”, instead of full spelling.

L44/Table 1 footnote. Change “lincospectin” (more like a commercial product that is to be avoided if possible for a publication) to two agents “the lincomycin and spectinomycin combination” which is seen later in L109 .

L46/48, etc. Would change “antibiotic” to “antimicrobial” for consistency as this ms uses mostly “antimicrobial”. Please the check the ms.

L50/L112/L166/L315, etc. Use Greek “β” for “beta”. Check the ms for consistency.

L52. Though “AMR” is introduced, the ms continues to write “antimicrobial resistance” (such as in L169, L193, L232, L292, L313 and L377, etc.

L73. “CLSI, 2023” needs to be provided by the reference (Ref 53; the order number has to be renamed) for clarity as there are vrous CLSI documents (M100, or VET01S, for example).

L121. Add a space after “32”.

L143. Write  “proteins” (plural).

L157. Write all “van” genes in italic.

L161. No need to reintroduce “COWT”.

L169. Would write uppercase first letters for CARD: Comp Anti…Resis… Data…

L196. Would write “Figures 4-6” for three separate cited figures.

L215/L216. Write lowercase italic genes “van” and “ant(6)-la”.

Table 1/Figure 1. The order of antimicrobial agents could be redone, either alphabetically or place related agents together, β-lactams (ampicillin and ceftiofur), gentamicin, spectinomycin, lincomycin, trylosin, etc.

L333/L497. Not italicize “spp.”

L415/L446, etc. Refs, 4, 15, 19, 20, etc. Use lower case article title. Chenck other refs.

L443/Ref 14, Italicize “vanA”.

Comments on the Quality of English Language

Moderate revisions are helpful.

Round 2

Reviewer 1 Report

Comments and Suggestions for Authors

Good for publication